# Expression Quantitative Trait Locus of Wood Formation-Related Genes in *Salix suchowensis*

**DOI:** 10.3390/ijms25010247

**Published:** 2023-12-23

**Authors:** Li Chen, Liyan Liu, Guo Yang, Xiaoping Li, Xiaogang Dai, Liangjiao Xue, Tongming Yin

**Affiliations:** State Key Laboratory of Tree Genetics and Breeding, Jiangsu Key Laboratory for Poplar Germplasm Enhancement and Variety Improvement, Co-Innovation Center for Sustainable Forestry in Southern China, Nanjing Forestry University, Nanjing 210037, China

**Keywords:** *Salix suchowensis*, eQTL, wood formation, regulatory networks

## Abstract

Shrub willows are widely planted for landscaping, soil remediation, and biomass production, due to their rapid growth rates. Identification of regulatory genes in wood formation would provide clues for genetic engineering of willows for improved growth traits on marginal lands. Here, we conducted an expression quantitative trait locus (eQTL) analysis, using a full sibling F_1_ population of *Salix suchowensis*, to explore the genetic mechanisms underlying wood formation. Based on variants identified from simplified genome sequencing and gene expression data from RNA sequencing, 16,487 eQTL blocks controlling 5505 genes were identified, including 2148 cis-eQTLs and 16,480 trans-eQTLs. eQTL hotspots were identified, based on eQTL frequency in genomic windows, revealing one hotspot controlling genes involved in wood formation regulation. Regulatory networks were further constructed, resulting in the identification of key regulatory genes, including three transcription factors (*JAZ1*, *HAT22*, *MYB36*) and *CLV1*, *BAM1*, *CYCB2;4*, *CDKB2;1*, associated with the proliferation and differentiation activity of cambium cells. The enrichment of genes in plant hormone pathways indicates their critical roles in the regulation of wood formation. Our analyses provide a significant groundwork for a comprehensive understanding of the regulatory network of wood formation in *S*. *suchowensis*.

## 1. Introduction

The requirement to reduce the emission of greenhouse gas and use more bioenergy, rather than relying on fossil fuels, has increased interest in bioenergy crop cultivation. Willow (*Salix* spp.) is considered as a short-rotation coppice (SRC) for the bioenergy, biofuel and bioproduct industries [1,2,3]. Meanwhile, willows are considered a highly promising tree species for soil remediation, due to their exceptional ability to hyperaccumulate soil pollutants such as petroleum hydrocarbons, polychlorinated biphenyls, and cadmium [4,5,6,7]. One of the major tasks of willow breeding is to cultivate varieties with high biomass. Wood is the primary form of biomass accumulation in woody plants like willow. Hence, gaining insights into the regulatory mechanisms of wood formation is a pivotal step in identifying driver genes controlling wood formation and plant growth, which can be applied to guiding breeding approaches in willow.

Wood (secondary xylem) is generated through the inward activity of vascular cambium. Xylogenesis is an integrative dynamic process that involves cell division, cell expansion, secondary cell wall (SCW) thickening and programmed cell death (PCD) [8,9,10]. In Arabidopsis, significant advancements have been made to understand the molecular mechanisms of wood biosynthesis and xylem development [11,12,13]. Arabidopsis has been shown to undergo stem swelling similar to secondary growth, when treated with decapitation or exposed to a short-day environment [14]. By comparing the gene expression profiles of induced secondary xylem with those of normal tissue using microarray technology, numerous differentially expressed genes have been discovered [15]. According to Ko et al. [16], there was clear differential expression observed in 700 genes during the transition from primary growth to secondary growth in Arabidopsis. Furthermore, several auxin-modulating genes, such as Auxin/Indole acetic acid repressors (*Aux/IAA*), auxin response factor (*ARF*), and auxin efflux carrier genes, have been reported in the induction of secondary growth in the stem. The gene expression profiles of different isolated tissues indicate several genes belonging to transcription factors (TFs), such as G2-like, NAC, AP2 and MYB TF families, involved in regulating phloem-cambium cells [17]. As previously noted, Arabidopsis is an excellent model for studying the genomics of secondary growth. However, it is quite difficult to comprehensively dissect the genomic mechanisms of the wood formation using this system, due to the considerable differences between herbaceous induced secondary growth and woody natural secondary growth. Schrader et al. [18] identified three genes (CLAVATA1-like, (*PttCLV1*), CLAVATA1-like (*PttRLK*) and AINTEGUMENTA-like (*PttANT*)), which showed specific expression in the vascular layer through the gene expression map in poplar wood-forming tissue. In another study, 23 genes that were most likely to encode monolignol biosynthesis enzymes were identified, of which 18 genes showed specific expression in developing xylem tissues [19]. Overexpression of *Ptr-miR397a*, targeting the laccase (*LAC*) genes, resulted in a reduction in lignin content in transgenic poplar trees, suggesting the crucial role of *LAC* in lignin synthesis [20]. These studies provide a range of genetic evidence for a comprehensive understanding of the secondary growth process.

Wood is composed of SCWs, with its three major components being cellulose, hemicellulose and lignin. Similar to many cellular development and differentiation processes, the deposition of SCWs requires the coordinated expression of SCW biosynthesis genes, which is controlled by hierarchical transcriptional networks [21,22]. The NAC TFs serve as the top-level master switches in the transcriptional networks, while the MYB TFs act as the second-level regulators. Together, they coordinately regulate the differentiation and biosynthesis of SCW [23,24,25,26]. In Arabidopsis, a number of NAC domain proteins, such as vascular-related NAC domain 6 (*VND6*), NAC secondary wall thickening promoting factor 1 (*NST1*), and secondary wall-associated NAC domain 1 (*SND1*), have been proposed to induce indirect expression of other TFs and genes related to cell wall synthesis [24,27,28,29]. The homologs of master switches have been functionally characterized, and named wood-associated NAC domain transcription factors (WNDs) in woody plants [30,31,32]. It has been demonstrated that overexpression of *PtrWNDs* can cause ectopic deposition of wood components [31,33]. The two *PtrWND1B* isoforms have also been shown to regulate fiber cell-wall thickening during wood development [34]. Furthermore, Lin [35] reported the reciprocal cross-regulation of *PtrVND* and *PtrSND*. WNDs activate the transcriptional regulatory network by binding to the secondary wall NAC-binding element (SNBE) in the promoter of downstream target genes [33]. *PtrMYB2/21* and *PtrMYB3/20*, the homologs of *AtMYB46* and *AtMYB83* in poplar, have been found to activate the biosynthesis pathways of lignin, cellulose and hemicellulose [36,37]. The available evidence suggests that *PtrMYB2/3/20/21* are direct targets of *PtrWNDs* [31,33]. Homologs of *PtrMYB2/3/20/21* from other tree species, such as eucalyptus and pine, also activate the entire SCW biosynthesis process [38,39]. On the other hand, a battery of TFs has been found to play a negative role in wood formation. Lignin biosynthesis-associated transcription factor 1 (*LTF1*), one of the homologs of *AtMYB4*, specifically represses lignin biosynthesis by binding to the promoter of a key lignin biosynthetic gene (4-coumarate:CoA ligase (*4CL*)) [40]. These findings indicate that wood formation is a complex process, controlled by multiple molecular networks.

Gene expression is an important molecular phenotype that associates genetic variation with organismic phenotype. The role of gene expression changes in phenotypic variation; species evolution has long been postulated, and has been supported by recent studies [41,42,43]. The advent of expression quantitative trait locus (eQTL) mapping, a powerful approach, has shed light on gene regulatory relationships. With the rapid advancements in sequencing technologies and substantial reduction in cost, genome and transcriptome analyses have become more accessible, providing a solid foundation for investigating the genetic properties of gene expression variation. Genome-wide eQTL mapping has been employed in numerous plant species to identify candidate genes responsible for important traits such as grain size [44], kernel oil [45], flowering time [46] and anthocyanin content [47]. In this study, we aimed to explore the genetic regulatory networks underlying wood development by integrating sequence diversity and gene expression data.

Previous studies have laid a solid groundwork for understanding the process of secondary xylem development in woody plants. However, our understanding of the genetic regulatory networks involved in wood formation remains incomplete. Specifically, limited research has been conducted on the genetic background of wood formation in willow. *Salix suchowensis*, a fast-growing shrub willow showing ease of vegetative propagation, presents an advantageous model for studying wood development. Furthermore, the high-quality chromosome-scale genome of *S. suchowensis* was released in 2020 [48]. In our investigation, we performed eQTL mapping, using a full-sib F_1_ family population of *S. suchowensis* [49]. Our objectives were as follows: (1) to identify a gene set comprising wood formation-related genes (WFRGs) in *S. suchowensis*; (2) to predict candidate master regulators involved in wood formation; and (3) to reveal the gene regulatory networks governing the process of wood development. These findings will contribute to our understanding of the impact of genetic variants on wood development in woody plants and provide clues for breeding woody plants with improved growth traits.

## 2. Results

### 2.1. Gene Expression Variability in the Transcriptome Dataset

A general analysis of transcriptome data revealed that 89.08% (32,904) of genes in the *S. suchowensis* genome exhibited expression levels with FPKM > 0. On average, 29,304 expressed genes were detected in each individual. Hierarchical clustering showed significant differences in gene expression between offspring and parents, indicating that a large number of offspring might have undergone transgressive inheritance (Figure 1B). Applying the criteria described in the methods, a total of 2805 DEGs were identified, comprising 1353 (48.24%) up-regulated genes and 1452 (51.76%) down-regulated genes in NF2, compared to LS7. The volcano plot (Figure 1C) highlights all up- and down-regulated DEGs. Further analysis using Gene Ontology (GO) terms revealed a notable enrichment of DEGs in plant hormone-related pathways (Figure 1D). Plant hormone-related genes exhibit significant differential expression between parents (Appendix A), indicating obvious differences in plant hormone synthesis and metabolism among parents of *S. suchowensis*.

### 2.2. Identification of SNPs and the Genomic Location of Variants

After applying quality control filters, we identified 169,699 high-confidence SNPs from 458.67 Gb of simplified genome sequencing. We observed that these markers were particularly densely distributed on Chr13 and Chr15 (Appendix A). More than half of SNPs (103,941, 61.25%) were localized in the upstream or downstream region of genes, where the main binding sites of regulatory elements are. Additionally, 16.98% of the SNPs (28,823) were found in intergenic regions, while only 8.54% of the SNPs (14,486) were distributed within coding sequences, including 7781 synonymous and 6705 nonsynonymous SNPs. (Appendix A). Notably, among these SNPs, 629 were classified as large-effect SNPs occurring in the coding sequences of 596 genes. These large-effect SNPs may result in frameshift variants, premature stop codons, disruptive splice variants, and other significant alterations. Some of these genes, such as *4CL1*, *LAC6* and agamous-like MADS-box (*AGL21*), have been well studied in model plants. It is likely that these genes play crucial roles in the growth of *S. suchowensis*, and warrant further investigation and attention.

### 2.3. Global eQTL Mapping-Linked Variations and Gene Expression

After filtering out genes with low expression levels, we obtained a total of 25,197 genes for eQTL mapping. At a rigorous Bonferroni-corrected α = 0.05 (*p* = 1.2 × 10^−11^), 110,517 SNPs were significantly associated with the 5517 genes. By selecting lead SNPs within a 10 kb interval that were significantly associated with the same gene, we obtained a total of 16,487 eQTL blocks targeting 5505 genes. Among these genes, the majority (4866, 88.39%) had more than 5 eQTLs, while only 234 (4.25%) genes had a single eQTL (Figure 2C). This result indicates that the expression variation of most genes in *S. suchowensis* is under complex genetic control. Analyzing the relative distance from the eQTLs to their target genes, we found 2148 (11.53%) cis-eQTLs and 16,485 (88.47%) trans-eQTLs associated with 1737 and 5490 genes, respectively. As expected, a strong enrichment along the diagonal was observed when plotting the positions of significant eQTLs, relative to their target genes (Figure 2A). This observation suggests that the majority of genes are subject to local regulation. Furthermore, the overall *p*-value of cis-eQTLs was found to be higher than that of trans-QTLs (Wilcoxon rank sum test, *p* = 3.01 × 10^−8^), indicating that the cis-eQTLs exerted a stronger influence on determining expression variation, compared to trans-eQTLs (Figure 2B).

### 2.4. Expression Characteristics of WFRGs

In *S. suchowensis*, we have identified 878 genes that are involved in wood formation. These genes were categorized into four major categories (cell elongation, SCW biosynthesis, TFs, and phytohormones), and further divided into 10 subclasses, based on their functions and biological processes (Appendix A). To analyze the gene expression patterns during wood development in *S. suchowensis*, we incorporated two publicly available datasets. The transcriptome data revealed that 770~837 genes were expressed, accounting for 89.09% to 85% of the WFRGs. We conducted DEG analysis on four datasets, and found that the parents, extreme growth rate, tension wood, and opposite wood exhibited a differential expression of 50, 28, 76, and 89 WFRGs, respectively (Figure 3A). These DEGs are likely to play crucial roles in xylem development. Interestingly, more than 50% of the differentially expressed WFRGs were found to be involved in the biosynthesis of lignin, cellulose, and pectin (Figure 3B–E). This suggests that SCW biosynthesis and modification are key factors influencing the growth rate and wood quality.

### 2.5. Identification of eQTL Hotspots, Their Functional Annotation and Effect on S. suchowensis Wood Traits 

As reported in previous studies [50,51], eQTL hotspots refer to genomic regions that control the expression of multiple genes and have the ability to modulate metabolic pathways. A permutation test indicated that a given window needed to harbor at least 81 eQTLs to be identified as an eQTL hotspot. In total, 194 eQTL hotspots were identified across different chromosomes, including Chr2, Chr5, Chr10, Chr13, Chr14, Chr15, and Chr16 in the *S. suchowensis* genome (Figure 4). Since some of these hotspots were in close proximity to each other, we examined the intervals between them, and merged the eQTL hotspot regions, resulting in 12 consolidated hotspots (Appendix A). A total of 1226 genes were regulated by the above 12 hotspots, and GO analysis showed enrichment in cell wall organization and biogenesis, phytohormone-mediated signaling and metabolic processes (Appendix A). Notably, when investigating the target genes associated with the phytohormone pathway, we found that they included well-known classical genes involved in auxin response (Table 1, Appendix A). In particular, Hot 12 on chromosome 16 exhibited eQTLs for three genes involved in cell wall biosynthesis. Through the pipeline described in the Method section, we identified potential trans regulators, including cellulose synthesis-associated glycosyltransferases 1 (*CAGE1*), fasciclin-like arabinogalactan 13 (*FLA13*), STRUBBELIG-receptor family 6 (*SRF6*), polygalacturonase clade family 11 (*PGF11*) and caffeoyl shikimate esterase (*CSE*), which are associated with phytohormone transmission and cell wall metabolic pathways. Among these candidate regulators, some have been previously reported to be involved in wood formation, such as *CSE*, *CAGE1* and *FLA13*, while others, such as *SRF6*, represent novel genes with potential roles in this process [52,53,54,55,56]. These findings provide a strong basis for further validation.

### 2.6. Regulatory Networks Provide a Comprehensive View of Wood Formation

eQTL analysis showed that 11.5% (101) of WFRGs were regulated by at least one eQTL. Among them, 66.3% (67) of the genes were regulated solely by trans-eQTLs, while 33.7% (34) of the genes were regulated by both cis- and trans-eQTLs. Overall, the WFRGs exhibited widespread expression in our samples, with approximately one-eighth of them being regulated by cis- and/or trans-eQTLs. 

After a series of screening, we identified 214 pairs of regulatory relationships, which included 115 regulatory factors and 26 target genes. The regulatory factors were classified based on existing knowledge, with a focus on known regulatory genes, such as hormones and TFs involved in the secondary growth of *S. suchowensis* (Appendix A). Further analysis revealed that each regulatory node controlled 1~4 target genes, while each target gene node was regulated by 1~27 regulatory factors. To construct the network, we selected regulatory relationships with a minimum of five nodes. The resulting network, depicted in Figure 5, consisted of four regulatory networks named network A, B, C, and D. We highlighted WFRGs in orange, and observed their involvement in the biosynthesis, modification, and degradation of SCW in the regulatory network. Networks A and B demonstrate that the lignin pathway is regulated by several protein kinases (PKs) and TFs. Cellulose Synthase-like D3 (*CSLD3*) and β-galactosidase 17 (*BGAL17b*) were identified as the main targets in network C, indicating that the deposition of SCW is regulated through the coordinated control of multiple biological processes [57,58,59]. Network D focused on glucose metabolism during SCW formation, and included three key genes, namely *BAM1*, *HAT22*, and *MYB36* [46,60,61,62,63].

### 2.7. Potential Regulators and Their Properties in Wood Formation

As shown in Appendix A, the regulators were categorized into six groups, based on their functions, including WFRGs, TFs and transcript regulators (TRs). Interestingly, we found four genes (*CLV1*, Cyclin-dependent protein kinase regulator *2;4*, (*CYCB2;4*), Cyclin-dependent kinase B2;1 (*CDKB2;1*), and BARELY ANY MERISTEM 1, (*BAM1*)) among the regulators that are potentially involved in cambium cell division and differentiation [64,65]. Furthermore, three TFs (Jasmonate zim-domain 1 (*JAZ1*), *HAT22*, and *MYB36*) were found to be closely associated with the synthesis and metabolism of SCW. Gene expression analysis revealed a strong correlation between these genes and their target genes (Figure 6A,D). 

The gene expression profiles of *S. suchowensis* showed that *BAM1*, *HAT22* and *JAZ1* exhibited high expression levels throughout the entire growth period, with significant temporal fluctuations (Figure 6C). When comparing *S. suchowensis* with different growth rates, we observed significant differences in the expression of *BAM1* and *HAT22* at t4 (195 days after planting) (Appendix A). At this time, *BAM1* was highly expressed in the slow-growing clone, while *HAT22* was highly expressed in the fast-growing clone (Figure 6D). Previous studies have demonstrated that *BAM1* not only drives the division of cambial cells in the root of Arabidopsis, but also is required for root lignification [65,66]. Based on the expression characteristics of *BAM1*, it is inferred that *BAM1* may promote lignification in *S. suchowensis*, leading to premature cessation of cell division in branches and thereby resulting in a phenotype of low height and high diameter. Overexpression of *PtrHAT22* is associated with a decrease in lignin and cellulose content, a thinning of SCW, reduced growth, and an increase in hemicellulose content [62]. At t4 stage, *HAT22* exhibits high expression levels in fast-growing clones, suggesting that this gene may also be involved in the regulation of wood formation in *S. suchowensis*. However, the regulatory mechanism may be slightly different from that of poplar, and further research is needed to elucidate this. At t1 (45 days after planting), *JAZ1* exhibits high expression levels in both genotypes, indicating its potential role in mediating IAA to promote initial growth. *JAZ1* showed high expression during the rapid growth period (t3, 135 days after planting) and the slow growth period (t5, 240 days after planting) (Figure 6D and Appendix A). Based on previous studies [67], we hypothesize that *JAZ1* may mediate different hormones at these two stages, to regulate wood formation. In addition, we found that during the early stage (t1~t4) of *S. suchowensis*, *CLV1* was predominantly highly expressed in the slow-growing clone, suggesting that *CLV1* may play a role in inhibiting wood growth.

Some hormone synthesis and transduction genes are also involved in regulatory networks, such as *IAA27*, ethylene receptor 1 (*ETR1*), basic pentacysteine 6 (*BPC6*), serine carboxypeptidase 28 (*SCP28*) and *MYBH*, and they have a strong expression correlation with their targets (Figure 6A,B). Our results show that five hormones (IAA, JA, ABA, BR, ETH) mainly regulate wood formation in *S. suchowensis*. Previous studies have shown that IAA, JA and ETH play a role in regulating cambium stem-cell activity, vascular-cambium cell differentiation, and interfascicular cambial activity, respectively [68]. Recent studies have emphasized the control of ABA and BR on the secondary growth of trees [69,70]. Our understanding of the regulatory mechanism of hormones on secondary growth is still limited, and these results provide important clues for a deeper understanding of the regulatory mechanism of wood information.

Interestingly, there was not only mutual feedback, but also self-feedback, regulation among the three genes (*BGAL17b*, cinnamoyl-CoAreductase 2a (*CCR2a*) and *CCR2b*), in above networks (Figure 5). We found that *BGAL17b* is regulated by a cis-eQTL located 1354 bp upstream of the gene (Figure 6D). Similarly, we observed the eQTL regulation of two *CCRs*, and found that they were located in the adjacent region (<100 Kb) of Chr 2 (*CCR2b:* Chr2: 222,270~224,376; *CRR2a*: Chr2: 277,848~279,954), and were regulated by three eQTLs (Chr2: 222,099; 238,052; 339,848) simultaneously. In addition, *CCR2b* was also regulated by an eQTL, which was located in 171 bp relative to the transcription initial site; in other words, *CCR2b* was regulated by a cis-eQTL (Figure 6E). A large number of studies have demonstrated that the regulation of gene transcription by cis elements can lead to phenotypic variation [71,72], and we speculate that a similar mechanism may also exist in *S. suchowensis* wood formation.

## 3. Discussion

### 3.1. A Large Population Helps to Interpret the Rapid Growth of S. suchowensis

Complex quantitative traits, like wood formation, are controlled by multiple genes, making it challenging to understand their genetic basis. Constructing regulatory networks provides a foundation for comprehensively describing gene expression regulation across the entire genome, and is a crucial approach for unraveling complex biological processes and gene functions. With advancements in technology and reduced sequencing costs, new methods such as genome-wide association studies (GWAS) have emerged, to investigate interesting traits using high-throughput sequencing. While these analyses can identify genetic variations and phenotypic traits, detecting regulators that are distantly located from the target gene within the network remains challenging [73]. 

eQTL mapping aims to uncover the relationship between gene expression and genetic variation, and identify causal genes that control specific traits [74,75]. According to our knowledge, the diversity of gene expression is mostly caused by cis- and trans-regulation [76]. eQTLs are classified into cis and trans, based on their physical location relative to genes [77]. eQTL has been successfully applied to explore the genetic mechanisms of phenotypes, such as flowering time in Arabidopsis, chlorogenic acid content in poplar, and flesh color in sweet potatoes [51,78,79]. Similar to the above studies, which were conducted on 160, 917, and 84 samples, respectively, this project was also tested on a large genetic population (120 hybrid offspring). The genetic population not only ensures the stability of genetic information, but also provides a rich range of phenotypic variations, making it easier to analyze phenotypic variations. RNA-seq analysis revealed that 2805 genes were differentially expressed between parents. In particular, genes related to the synthesis and signaling of major hormones such as IAA, JA, CTK, ETH and ABA exhibited significant changes (fold change: 0.04~11.39) (Appendix A). Through eQTL mapping, we found 16,487 eQTLs that regulate the expression of 5505 genes. Further analysis revealed the coordinated regulation among several processes of wood formation.

### 3.2. Potential Regulators at Distant eQTL Hotspots

According to the definition of hotspots, we identified 12 eQTL hotspots that regulate a total of 1226 target genes. Previous studies have revealed that hotspots often contain key regulators that control important biological processes [80]. In our study, we observed that target genes form four eQTL hotspots (Hot4, Hot7, Hot8, and Hot12), which were significantly enriched in hormone signaling and cell wall assembly. Within these hotspots, we proposed five potential master regulators (*FLA13*, *CAGE1*, *SRF6*, *CSE*, and *PGF11*), most of which have been previously reported in Arabidopsis [53,55] and poplar [56,81,82,83]. For example, *FLA* encodes a fasciclin-like arabinogalactan protein, and has been implicated in GA-mediated tension wood formation [53]. Our results showed an association between *FLA13* and the auxin response factor (EVM002776, gretchen hagen 3 (*GH3*)), suggesting a potential synergistic role of IAA and GA in wood development. This hypothesis was further supported by recent research on hormones and vascular cambium activity [84]. In addition, we identified *CAGE1* as one of the potential master regulators. Previous studies have demonstrated that *CAGE1* promotes cellulose synthesis in SCW of Arabidopsis [56]. The expression of *CAGE1* was found to be higher in tension wood compared to normal wood, consistent with the phenotype of increased cellulose content in tension wood [85]. *CAGE1* also regulated the expression of *GH3*, indicating the potential role of auxin signaling in stimulating cellulose synthesis. Furthermore, our hotspot analysis detected a new regulatory factor, *SRF6*, which encodes a protein from STRUBBELIG-receptor family 6. In Arabidopsis, the atypical receptor-like kinase STRUBBELIG (*SUB*) is known to mediate organ formation and shape [52]. Recent studies have shown that SUB signaling also influences the biosynthesis of cellulose, a major component of the cell wall [54]. Therefore, we speculate that *SRF6* may indirectly control the properties of SCW, through the receptor kinase signaling pathway. The eQTL hotspot analysis identified five potential wood synthesis regulatory genes, of which four homologous genes have been reported to be involved in the regulation of wood formation in other species, suggesting that they may also be involved in the regulation of wood formation in *S. suchowensis*. Another gene, *SRF6*, is a less studied novel gene, and our study suggests that this gene may also play a regulatory role in wood formation.

### 3.3. eQTLs Linked Gene Expression with Genotype, Which Was Helpful for Mining Candidate Genes

As an effective analytical tool, we applied eQTL to explore potential regulatory factors in wood formation of *S. suchowensis*. By integrating the eQTL map of young stems with gene co-expression analysis, we have successfully identified various regulatory genes, including TFs, TRs, and hormone-related genes, which are likely associated with secondary growth (Appendix A). Among these, regulatory factors such as *CLV1*, *BAM1*, *CYCB2;4* and *CDKB2;1* are thought to regulate cell division and differentiation, while several genes like *HAT22*, *MYB36* and *JAZ1* are involved in wood formation, through the regulating of SCW synthesis. Studies in poplar have shown that the transcriptional activation of *CYCB* and *CDKB* is associated with cambium cell division during early spring [64]. Our results suggest that these genes may also be involved in the regulation of wood formation in *S. suchowensis*. *CLV1* and *BAM1*, encoding *CLV1*-related receptor kinases, are known to play crucial roles in the function of both root- and shoot-meristem function in Arabidopsis [65,66]. It is well known that *WUS*/*CLV* (*WUSCHEL/CLAVATA*) maintains and controls the balance between organogenesis and meristematic tissue. This stem-cell regulatory mechanism has been confirmed in shoot apical meristem (SAM) and root apical meristem (RAM), but research on this mechanism in vascular cambium (VCAM) is still rare [86]. Our results suggest that there may be a regulatory mechanism similar to *WUS/CLV* in VCAM. *JAZ1* has been identified as a suppressor of JA signaling, and recent studies have also highlighted its importance in the crosstalk between JA and IAA [67]. In this work, we found that both *IAA27* and *JAZ1* target *BGAL17b*, indicating their involvement in SCW generation and modification, and providing further evidence of the crosstalk between these two hormones. In poplar, overexpression of the bZip TF *HAT22* has been shown to reduce the contents of lignin and cellulose, as well as the thickness of SCW, leading to inhibited stem thickening [62]. *MYB36* has long been considered a crucial regulator for lignin polymerization, although most of the relevant studies have been conducted in the model plant Arabidopsis, and have primarily focused on its expression in the root system [87,88]. Our study suggests that *MYB36* may also mediate the lignification in *S. suchowensis*, and promote wood formation.

Currently, the application of eQTL is mainly focused on crops and horticultural plants, while research on trees, especially non-model species like *S. suchowensis*, is still limited. To the best of our knowledge, this study represents the first comprehensive eQTL analysis conducted on *S. suchowensis*, providing novel insights into the genetic regulation of complex traits in other woody plants. We investigated the regulatory networks involved in wood formation, using a systematic genetic approach that integrated eQTL mapping with gene co-expression. Combining this approach with GWAS and QTL can further identify candidate genes, and narrow down potential regions of interest. Furthermore, by combining eQTL mapping with haplotype analysis, we can uncover allele-specific variations that contribute to phenotypic differences [51,89,90]. Additionally, our dataset can be utilized to explore the genetic regulation of other traits in *S. suchowensis*, including secondary metabolite content and insect resistance.

## 4. Materials and Methods

### 4.1. Plant Material

Plant materials were all derived from a full-sib F_1_ family population, which was generated in our laboratory through the crossbreeding of female *S. suchowensis* NF2 with male *S. suchowensis* LS7, in a previous study [49]. Field measurements revealed that NF2 exhibited superior seedling height and stem diameter, compared to LS7. As shown in Figure 1A, we randomly collected 10 cm young stems of 120 different genotypes of the *S. suchowensis* for transcriptome sequencing. The samples were collected in May 2019, which was the stage with the highest growth rate of the *S. suchowensis* [91]. In order to distinguish the apical meristem from the lateral meristem, the stem was cut from a leaf branch at an angle of 30°. The sample was frozen in liquid nitrogen and stored at −80 °C, until used.

### 4.2. RNA Sequencing and Expression Profiling

Total RNA was isolated and purified from young stems using TRIzol reagent (Invitrogen, Carlsbad, CA, USA), following the manufacturer’s protocol. The integrity and amount of RNA in each sample was assessed using RNA Nano 6000 Assay Kit (Agilent Technologies, Santa Clara, CA, USA), and confirmed by electrophoresis on denaturing agarose gel. To capture mRNA from the total RNA, poly-T oligo-attached magnetic beads were used, and the captured mRNA was fragmented into small pieces at 94 °C. These small fragments of mRNA were then used to construct RNA libraries, following the manufacturer’s instructions (Illumina, San Diego, CA, USA). The libraries were sequenced using the 150 bp read option on Illumina Novaseq 6000 at Annoroad Gene Technology Corporation (Beijing, China). The raw reads underwent filtering using Trimmomatic v0.36 [92], to obtain clean data for further analysis. The filtered files were then mapped to the reference genome (*S. suchowensis* v2.0) [48], using STAR v2.7.9a [93] with default parameters. Read counts were generated from the resulting BAM files, using the featureCounts subroutine in the subreads package v.2.0.3 [94]. To quantify gene expression, normalized fragment per kilobase million (FPKM) values were calculated. PCA and sample clustering were performed using FPKM values, to identify any outlier samples. DEGs were identified using DESeq2 v1.32.2 [95] with a threshold of q-value < 0.05 and |log_2_(fold change)| > 1. In addition, RNA-Seq data for *S. suchowensis* at different developmental stages were downloaded from NCBI [96,97], and analyzed using the same pipeline.

### 4.3. SNP Calling of Population

In a previous study, we performed simplified genome sequencing on the mapping population described earlier, to identify variants associated with wood formation [98]. The sequencing data underwent several preprocessing steps. Firstly, low-quality data and adapter sequences were removed using Trimmomatic v0.36 [92]. Then, the clean reads were aligned to the reference genome [48] using BWA-mem v0.7.11 [99] with default parameters. Variants, including SNPs, short INDELs, and multiple nucleotide polymorphisms, were called using Freebayes v0.9.10 [100] with the parameter “–use-best-n-alleles 6”. In the following analysis, these variants were collectively referred to as SNPs, for simplicity. SNPs with a quality lower than 50 were removed, to ensure the accuracy of subsequent analysis. Population-level screening of SNPs was performed using an in-house Python script. Only SNPs with minor allele frequency (MAF) > 5% and missing rate < 10% were maintained for eQTL mapping. SnpEff v4.3t [101] was used to further analyze the genomic distribution and functional effects of remaining SNPs.

### 4.4. Construction of Homologous Gene Sets

Wood is a key factor in determining the economic value of *S. suchowensis.* Genes involved in wood formation, known as wood formation-related genes (WFRGs), are highly conserved in vascular plants, and have been extensively studied in poplar. To identify genes potentially involved in wood development in *S. suchowensis*, we performed a search for homologous genes using Orthofinder v2.5.4 [102], based on the gene annotation of *Populus trichocarpa* v3.0 [10,19,20,57,103]. In addition, we also identified TFs that are known to play a role in wood formation, based on previous studies. To further analyze the functions of WFRGs, we classified them into different categories, based on the biological processes they are involved in, using in-house Python scripts. Similarly, we constructed a gene set for hormone synthesis and hormone response, using the same methodology.

### 4.5. Identification and Analysis of eQTL

A gene was considered expressed if its FPKM was greater than 1 in more than 20% of the individuals, otherwise it was excluded. The expression values of the remaining genes were normalized using the qqnorm function in R. We utilized the computationally efficient MatrixeQTL package v2.1.0 [104] to perform the associations for SNP-gene pairs. A suggestive *p* value threshold of 1.2 × 10^−11^ (multiple test) was applied to reduce false-positive associations. To reduce the redundancy of eQTLs for certain genes, we grouped the significant SNPs into clusters, within a maximum distance of 10 Kb. Only clusters that contained at least three significant SNPs were considered as an eQTL block. The SNPs with the minimum *p* value in each eQTL block were chosen as markers for the locus for further analysis, although they may not necessarily be causal variants. To classify the mapped eQTLs as cis- and trans-eQTL, we defined a 10 Kb region surrounding each gene as the association region. If an eQTL was located within 10 Kb region of its target gene, it was classified as a cis-eQTL. Conversely, if the eQTL was located outside the 10 Kb region, it was classified as a trans-eQTL. 

To identify potential distant eQTL hotspots, we conducted a sliding window analysis with 1 Mb windows and 10 Kb steps, to count the number of trans-eQTLs along the genome. We performed 1000 permutation tests, to determine the threshold for declaring significant distant eQTL hotspots (*p* < 0.01). Any overlapping or adjacent hotspots that were likely to correspond with a single hotspot were merged together. Subsequently, we investigated the genomic distribution of target genes regulated by these hotspots, which could be associated with specific biological processes. To assess the functional enrichment of these target genes, we performed Gene Ontology (GO) enrichment analysis, using clusterProfiler [105]. Furthermore, we annotated gene functions using several public protein databases, including Swiss-Prot, GO and KEGG databases. We visualized the eQTL hotspots in the genome using Circlize v0.4.11 [106]. By integrating information on the genetic position of eQTL hotspots, gene annotation, and co-expression patterns among regulators, we identified potential master regulators for the enriched pathways. The criteria for determining the validity of regulatory relationships are summarized as follows: (1) regulatory factors should be located near significant eQTLs of target genes; (2) the Pearson correlation coefficient (PCC) value between regulatory factors and target genes must be greater than 0.4 or less than −0.4; and (3) the target gene and regulatory factor are differentially expressed simultaneously in at least one transcriptome (regulatory genes differentially expressed at 1.2 times or more, and target genes differentially expressed at 1.5 times or more).

### 4.6. Construction of Regulatory Networks for Wood Formation

The genomic bins (100 Kb) were grouped to identify genomic regions controlling no less than 2 WFRGs. Regulatory genes in genomic bins were firstly screened based on expression patterns. Both the obtained regulators and the target genes were applied to search the regulatory matrix from eQTL analysis, to extract regulatory pairs. The construction of the regulatory network between the identified regulators and targets was performed using Cytoscape v3.8.2 [107], with the edge-weighted force-directed layout method. To facilitate the classification of identified regulators, we employed iTAK v1.7a [108] to identify TFs, TRs and PKs in the whole genome. The candidate regulators were further classified, based on their functions.

## 5. Conclusions

This study focused on an F_1_ hybrid population of *S. suchowensis* to investigate the regulation of wood formation through eQTL analysis. By integrating genetic variation and gene expression data, we identified a total of 2148 cis-eQTLs and 16,485 trans-eQTLs across the genome. A total of 13 eQTL hotspots were detected, with four of them referred to as specific hotspots, as they are associated with cell-wall tissue development and plant hormone signaling. Analyzing the above regions, we identified five potential trans regulatory factors. Using a public database, a regulatory analysis of WFRGS was conducted, and genetic regulatory networks for wood formation were established. Through gene expression patterns, four genes (*CLV1*, *BAM1*, *CYCB2;4*, and *CDKB2;1*), which may regulate the division and differentiation of cambium cells and three key transcription factors (*JAZ1*, *HAT22*, and *MYB36*) were unearthed. These candidate genes provide a valuable direction and genetic resource for future targeted improvements in wood quality.

## Figures and Tables

**Figure 1 ijms-25-00247-f001:**
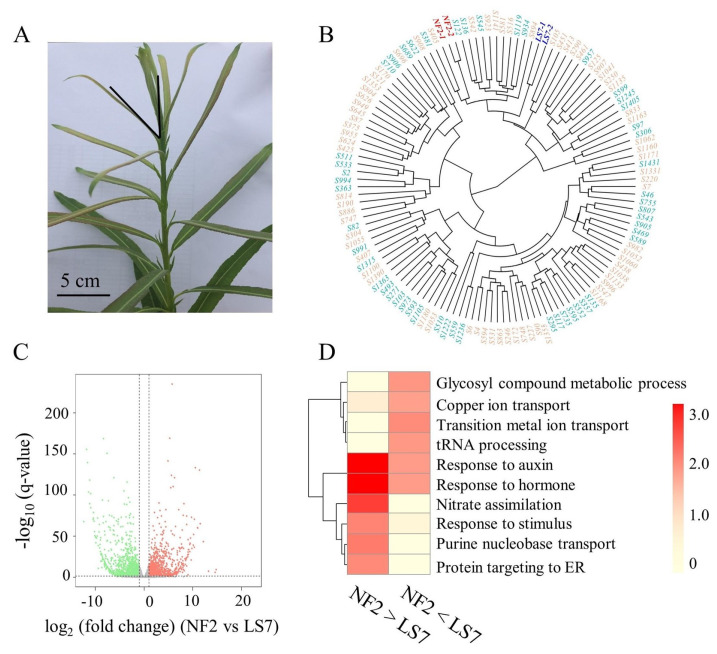
Sampling of *S. suchowensis* stem and parental transcriptome analysis. (**A**) A 10 cm stem segment. (**B**) Sample hierarchical clustering, based on gene expression. The blue, red, orange, and cyan fonts represent paternal individuals, maternal individuals, female offsprings, and male offsprings, respectively. (**C**) Volcano plot of whole-genome transcriptome of the parents. The x-axis represents log_2_(fold change), and the y-axis represents −log_10_(q-value). Orange dots represent up-regulated genes, green dots represent down-regulated genes, and genes with no significant changes are shown in gray dots. The criteria for identifying up-regulated and down-regulated genes are based on log_2_(fold change) > 1 and q-value < 0.05. (**D**) DEGs are enriched in hormone-related biological pathways. The numbers represent the significance (−log_10_(*p*-value)) of enrichment in this pathway.

**Figure 2 ijms-25-00247-f002:**
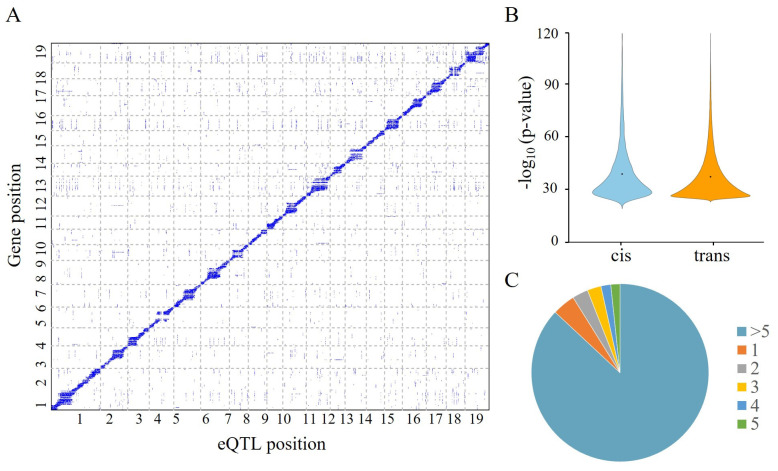
Identification of eQTL using RNA-Seq data in wood formation of *S. suchowensis*. (**A**) Genomic distribution of eQTLs and their targets. The x-axis shows the positions of eQTLs, while the y-axis represents the mapping of target genes. The color of each point indicates the associated *p*-value, where *p*-value < 10^−15^ is represented in gray, and the remaining values are represented in blue. (**B**) Violin plot displays the expression variation explained by cis- and trans-eQTLs, with black dots indicating the median of the distribution. (**C**) Number of eQTLs mapped for each gene.

**Figure 3 ijms-25-00247-f003:**
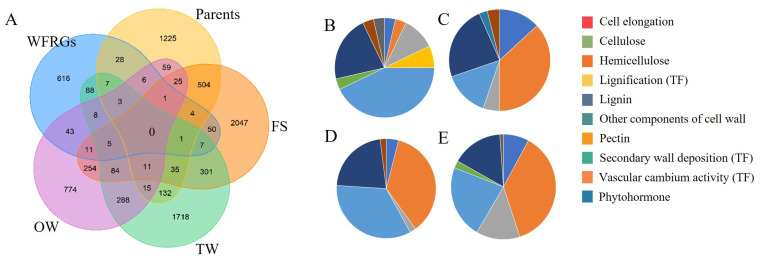
Expression characteristics of wood formation-related genes (WFRGs). (**A**) Venn diagram shows the DEGs identified from transcriptome datasets. The blue, purple, green, orange, and yellow regions represent WFRGs, DEGs in opposite wood compared to normal wood, DEGs in tension wood compared to normal wood, DEGs in two different growth rates of *S. suchowensis*, and DEGs in parents. The pie charts display the functional classification of differentially expressed WFRGs in different growth rates (**B**), opposite wood compared to normal wood (**C**), a comparison between two parents (**D**) and tension wood compared to normal wood (**E**).

**Figure 4 ijms-25-00247-f004:**
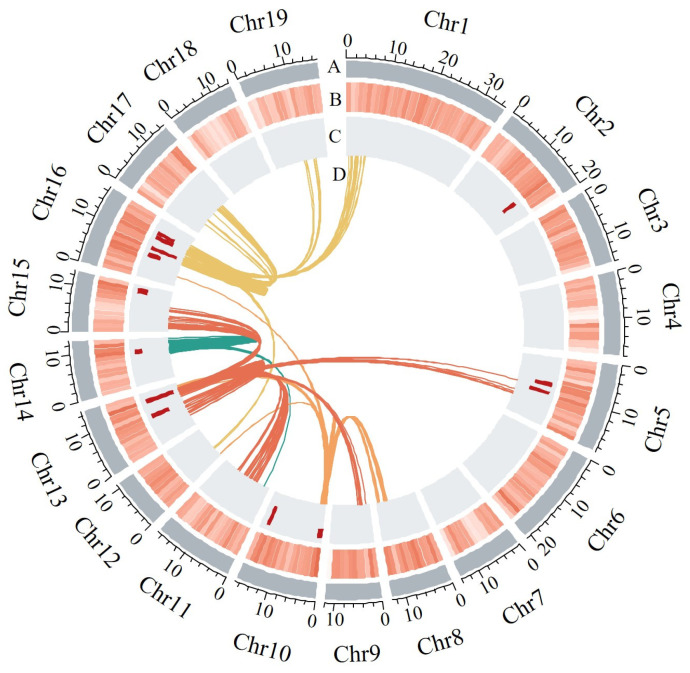
Visualization of eQTL hotspots in genome. (A) Nineteen chromosomes of *S. suchowensis* (Mb). (B) Counts of target genes in 1 Mb sliding windows across the genome. (C) Histogram showing the number of targets in 13 hotspots. (D) Links between four specific hotspots and their targets.

**Figure 5 ijms-25-00247-f005:**
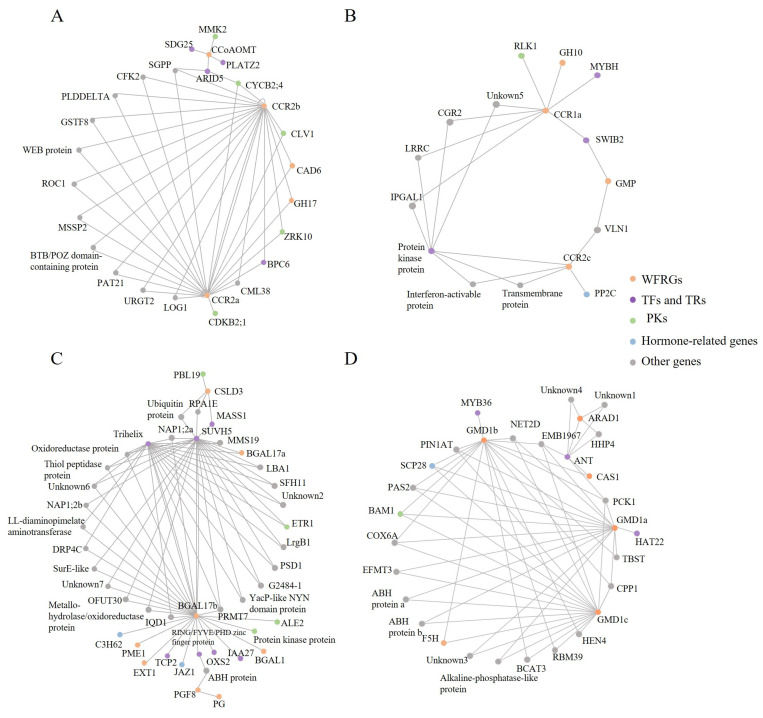
Regulatory networks governing wood formation in *S. suchowensis.* Regulatory networks controlling lignin biosynthesis (**A**,**B**), SCW deposition (**C**), and polysaccharide biosynthesis (**D**) during wood formation.

**Figure 6 ijms-25-00247-f006:**
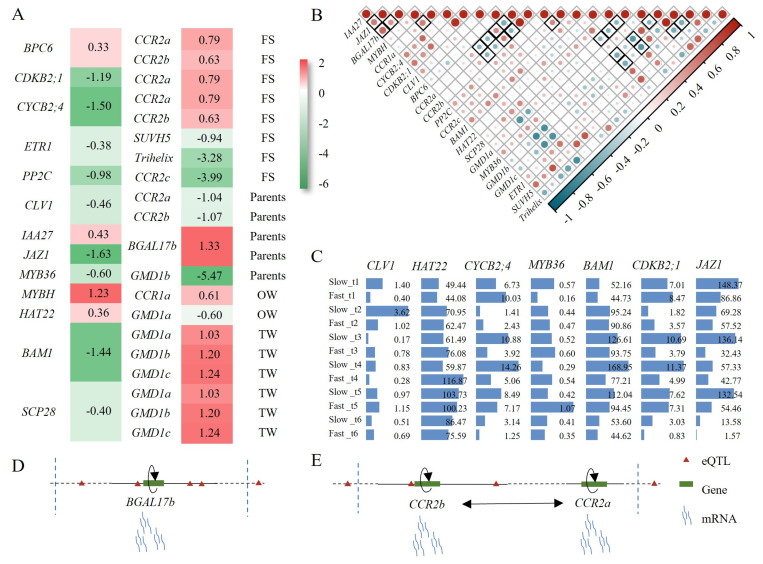
Analysis of potential key regulatory factors for wood information. Two heatmaps show the log_2_(fold change) (**A**) and Pearson correlation coefficients (**B**) of the regulatory factors and their target genes. The four RNA-seq datasets are abbreviated as FS, parents, OW, and TW, with their respective fold change values being fast vs. slow, female vs. male, tension wood vs. normal wood, and opposite wood vs. normal wood. The labels following the heatmap a represent the transcriptome datasets where gene differential expression occurred. (**C**) Expression profile of candidate regulatory factors during a full growth cycle. Regulation of eQTLs within a 100 Kb range near *BGAL17b* (**D**) and *CCRs* (**E**). Black arrow lines indicate the eQTL regulation of genes.

**Table 1 ijms-25-00247-t001:** Overview of targets and candidate regulators for specific pathways.

Hotspot ID	Chr	Hotspot Start	Hotspot End	Enriched Special Pathway	Target(s) Involved in Special Pathway	Candidate Regulators
Hot4	Chr10	0	1,410,000	hormone-mediated signaling pathway	EVM0000387 (*GH3.10*)	-
Hot7	Chr13	11,810,000	13,040,000	hormone-mediated signaling pathway	EVM0000387 (*GH3.10*)	EVM0029643 (*FLA13*)
Hot8	Chr14	8,740,000	10,000,000	hormone-mediated signaling pathway	EVM0002776 (*GH3.6*), EVM0041547 (*ARF2*)	EVM0029357 (*CAGE1*)
Hot12	Chr16	11,110,000	12,390,000	plant-type cell wall assembly	EVM0036007 (*PME3*), EVM0034292 (*PME3*), EVM0005816 (*COBL2*)	EVM0011519 (*SRF6*), EVM0008575 (*PGF11*), EVM0008454 (*CSE*)

## Data Availability

The article contains the supporting data for its findings, and the raw transcriptome sequencing data from this study are openly accessible in the National Center for Biotechnology Information (NCBI) SRA database, identified by the BioProject ID: PRJNA999089.

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
