# Peer review of "Expression Quantitative Trait Locus of Wood Formation-Related Genes in Salix suchowensis"

_ijms, 2023, doi:10.3390/ijms25010247_

Round 1

Reviewer 1 Report

Comments and Suggestions for Authors

In this manuscript, the authors presented the eQTL analysis in Willow to identify regulatory genes controlling the expression patterns of wood formation-related genes. Regulatory networks were also constructed to integrate the results from eQTL and co-expression analysis. The results would be interesting for the field of wood formation regulation in woody plants.

There are still some concerns about the manuscript of this version.

1. The details about two parental clones of the hybrid population need to be provided. Why this combination was selected? Is there a difference in growth traits and wood formation between them?

2. The methods of regulatory network construction should be described in more detail. What kind of connections were included? And how the thresholds were determined for the connections. 

Minor points:

1. The full names of genes should be listed where they are mentioned for the first time.

2. More information is required in figure legends. For example, in Figure 2. The title “Global eQTL mapping” is too brief. The title should cover all the contents in this figure.

3. Many software tools were applied in the analysis, the versions of some of them were listed, whereas the others were not. The version numbers need to be provided for all the tools. 

Comments on the Quality of English Language

 Minor editing of English language required

Author Response

Reviewer #1:

In this manuscript, the authors presented the eQTL analysis in Willow to identify regulatory genes controlling the expression patterns of wood formation-related genes. Regulatory networks were also constructed to integrate the results from eQTL and co-expression analysis. The results would be interesting for the field of wood formation regulation in woody plants. 

Response: Thanks for the comments. The questions are addressed point to point as below.

There are still some concerns about the manuscript of this version. 

Comments 1: The details about two parental clones of the hybrid population need to be provided. Why this combination was selected? Is there a difference in growth traits and wood formation between them?

Response 1: In the revision, the information of two parental clones are described in detail. They exhibit significant phenotypic differences. In previous field measurements, it was observed that the growth of maternal clone is better than paternal clone, with higher height and larger stem diameter. The differences in growth traits agree with significant differential gene expression between two clones.

Comments 2: The methods of regulatory network construction should be described in more detail. What kind of connections were included? And how the thresholds were determined for the connections.

Response 2: We rewrote the method section to provide more details about the approach for construction of the regulatory network. Thanks.

“The genomic bins (100 Kb) were grouped to identify genomic regions controlling no less than 2 WFRGs. Regulatory genes in genomic bins were firstly screened based expression patterns. Both the obtained regulators and the target genes were applied to search regulatory matrix from eQTL analysis to extract regulatory pairs.”

Minor points:

Comments 1:.The full names of genes should be listed where they are mentioned for the first time.

Response 1: Thank you for the suggestions. We have listed the full names of genes in the manuscript as recommended. Please refer to pages 2, 9, and 11 for details.

Comments 2: More information is required in figure legends. For example, in Figure 2. The title “Global eQTL mapping” is too brief. The title should cover all the contents in this figure.

Response 2: The title of Figure 2 has been revised to "Identification of eQTL using RNA-Seq data in wood formation of S. suchowensis." Additionally, the authors have made comprehensive modifications to the figure legends throughout the entire manuscript.

Comments 3: Many software tools were applied in the analysis, the versions of some of them were listed, whereas the others were not. The version numbers need to be provided for all the tools.

Response 3: During the revision, we included the versions of software as suggested. Please refer to the method section for details.

Reviewer 2 Report

Comments and Suggestions for Authors

The manuscript is an excellent attempt to understand the genetic and regulatory network of woof formation. Why authors did not validate the expression of some key candidate genes (through qPCR) identified with RNA-seq data. Kindly justify in the Methods section. 

Some of the comments for revisions are:

1. Page 2-   change "a number of genes with differential expression" to "numerous differentially expressed genes"

2. Page 2- rewrite for clarity "was clear biased expression observed" biased for what?

3. Page 2- change "As previously noted, Arabidopsis has been proven to be an excellent model" to " As previously noted, Arabidopsis is an excellent model"

4. Page 2- NAC and MYB TFs, as the top-level and second-level master switches in the transcriptional networks..should it have respectively to indicate one as top level and other as MYB? Kindly clear 

5. Page 3- genome of S. suchowensis was recently released in 2020 [..2020 is not recently..

There are some typographical mistakes, formatting/font issues as highlighted in the attached PDF.

Comments on the Quality of English Language

Minor improvement is needed.

Author Response

Reviewer #2: 

Comments 1: The manuscript is an excellent attempt to understand the genetic and regulatory network of woof formation. Why authors did not validate the expression of some key candidate genes (through qPCR) identified with RNA-seq data. Kindly justify in the Methods section.

Response 1: Thank you for the suggestion. In this study, eQTL analysis was performed to identify regulatory relationship among gene loci. The expression data were mainly used to screen key regulatory genes. Further validation of candidate genes using qPCR will be performed in subsequent studies.

Comments 2: Page 2-   change "a number of genes with differential expression" to "numerous differentially expressed genes"

Response 2: The sentence has been revised as suggested. Thanks.

Comments 3: Page 2- rewrite for clarity "was clear biased expression observed" biased for what?

Response 3: The sentence “was clear biased expression observed” was revised into “was clear differential expression observed”.

Comments 4: Page 2- change "As previously noted, Arabidopsis has been proven to be an excellent model" to " As previously noted, Arabidopsis is an excellent model"

Response 4: The sentence has been revised as suggested.

Comments 5: Page 2- NAC and MYB TFs, as the top-level and second-level master switches in the transcriptional networks..should it have respectively to indicate one as top level and other as MYB? Kindly clear

Response 5: Thanks for the suggestions. We rephrased the sentences as "The NAC TFs serve as the top-level master switches in the transcriptional networks, while the MYB TFs act as the second-level regulators. Together, they coordinately regulate the differentiation and biosynthesis SCW”.

Comments 6: Page 3- genome of S. suchowensis was recently released in 2020 [..2020 is not recently..

Response 6: The authors have removed the term "recently" in the revised version of the manuscript.

Reviewer 3 Report

Comments and Suggestions for Authors

This is a study investigate the regulation of wood formation through eQTL analysis by a F1 hybrid population of S. suchowensis.as result, four of the 13 discovered eQTL hotspots are the most associated with cell wall tissue development and plant hormone signaling. four genes (CLV1, BAM1, CYCB2;4, CDKB2;1) that may regulate the division and differentiation of cambium cells, and three key transcription factors (JAZ1, HAT22, MYB36) were unearthed.

This research has been well established and data are well presented. I recommend publishing after some minor writing issues are fixed and may also consider some suggestions below.

1.     The first paragraph describes how important of willow as a multi-use crop, however it did not discuss anything about “wood formation” in those uses. Just to the last sentence is not logic. Please further discuss why “Understanding the regulatory mechanisms of wood formation in willows is crucial for enhancing their biomass as biofuel and for soil remediation purposes.”

2.     Everyone read knows Arabidopsis has been well-studied:

Arabidopsis (Arabidopsis thaliana), the most well-studied herbaceous model species, has made significant advancements in molecular and genomic understanding of wood bio- synthesis and xylem development.

3.     It would be helpful simply make the second paragraph into a list table. (suggestion)

4.     Table 1, there different format of “,”. Also, different size of words.

5.     Italic all spp names

6.     Page 11, citations of the two species should be separate.

arabidopsis and poplar [53,55,56,81-83] -> arabidopsis [XX,XX] and poplar [XX,XX].
Potential regulators at distant eQTL Hotspots

7.     Any phenotype of the population work involved?

Comments on the Quality of English Language

It is really good. Only very few minor issues, like few spp names are not italic or format of "," inconsistant. 

Author Response

Reviewer #3:

This is a study investigate the regulation of wood formation through eQTL analysis by a F1 hybrid population of S. suchowensis.as result, four of the 13 discovered eQTL hotspots are the most associated with cell wall tissue development and plant hormone signaling. four genes (CLV1, BAM1, CYCB2;4, CDKB2;1) that may regulate the division and differentiation of cambium cells, and three key transcription factors (JAZ1, HAT22, MYB36) were unearthed.

This research has been well established and data are well presented. I recommend publishing after some minor writing issues are fixed and may also consider some suggestions below.

Comments 1: The first paragraph describes how important of willow as a multi-use crop, however it did not discuss anything about “wood formation” in those uses. Just to the last sentence is not logic. Please further discuss why “Understanding the regulatory mechanisms of wood formation in willows is crucial for enhancing their biomass as biofuel and for soil remediation purposes.”

Response 1: Thanks for the suggestions. We revised the first paragraph to focus on the importance of wood formation in plant growth and genetic breeding.

Comments 2:Everyone read knows Arabidopsis has been well-studied:

Arabidopsis (Arabidopsis thaliana), the most well-studied herbaceous model species, has made significant advancements in molecular and genomic understanding of wood bio- synthesis and xylem development.

Response 2: In the revision, we deleted the emphasis on the model species.

The rewrote sentence reads:

In Arabidopsis, significant advancements have been made to understand the molecular mechanisms of wood biosynthesis and xylem development.

Comments 3:It would be helpful simply make the second paragraph into a list table. (suggestion)

Response 3: Thanks for the suggestion. We created on table to list the discussed genes in the second paragraph.

Comments 4:Table 1, there different format of “,”. Also, different size of words.

Response 4: Done. We checked the symbols and font size in Table 1 and corrected the inconsistence in the formats.

Comments 5:Italic all spp names

Response 5: Done. We checked the spp names throughout the manuscript.

Comments 6:Page 11, citations of the two species should be separate.

arabidopsis and poplar [53,55,56,81-83] -> arabidopsis [XX,XX] and poplar [XX,XX].

Response 6: Thanks. The sentence has been revised to separate the citations, which now reads “... reported in arabidopsis [53,55] and poplar [56,81-83]”.

Comments 7:Any phenotype of the population work involved?

Response 7: In this manuscript, the results of eQTL analysis were reported, which belong to a large project including both phenotypic data and transcriptional data. The Phenotypic data have been reported earlier. We didn’t repeat this part in the current manuscript. We revised the method section to make the points clear.